# Vasari Scoring System in Discerning between Different Degrees of Glioma and IDH Status Prediction: A Possible Machine Learning Application?

**DOI:** 10.3390/jimaging9040075

**Published:** 2023-03-24

**Authors:** Laura Gemini, Mario Tortora, Pasqualina Giordano, Maria Evelina Prudente, Alessandro Villa, Ottavia Vargas, Maria Francesca Giugliano, Francesco Somma, Giulia Marchello, Carmela Chiaramonte, Marcella Gaetano, Federico Frio, Eugenio Di Giorgio, Alfredo D’Avino, Fabio Tortora, Vincenzo D’Agostino, Alberto Negro

**Affiliations:** 1Department of Advanced Biomedical Sciences, University “Federico II”, Via Pansini, 80131 Naples, Italy; 2Oncology Unit, Ospedale del Mare ASL NA1 Centro, Via Enrico Russo, 80147 Naples, Italy; 3Neuroradiology Unit, Ospedale del Mare ASL NA1 Centro, Via Enrico Russo, 80147 Naples, Italy; 4Neurosurgery Unit, Ospedale del Mare ASL NA1 Centro, Via Enrico Russo, 80147 Naples, Italy; 5Radiotherapy Unit, Ospedale del Mare ASL NA1 Centro, Via Enrico Russo, 80147 Naples, Italy; 6CNRS, Laboratoire J.A. Dieudonné, Inria, Universitè Côte d’Azur, Avenue Valrose, 06108 Nice, France; 7Nuclear Medicine Unit, Ospedale del Mare ASL NA1 Centro, Via Enrico Russo, 80147 Naples, Italy; 8Pathological Anatomy Unit, Ospedale del Mare ASL NA1 Centro, Via Enrico Russo, 80147 Naples, Italy

**Keywords:** magnetic resonance, glioma, VASARI, grade prediction, IDH

## Abstract

(1) The aim of our study is to evaluate the capacity of the Visually AcceSAble Rembrandt Images (VASARI) scoring system in discerning between the different degrees of glioma and Isocitrate Dehydrogenase (IDH) status predictions, with a possible application in machine learning. (2) A retrospective study was conducted on 126 patients with gliomas (M/F = 75/51; mean age: 55.30), from which we obtained their histological grade and molecular status. Each patient was analyzed with all 25 features of VASARI, blinded by two residents and three neuroradiologists. The interobserver agreement was assessed. A statistical analysis was conducted to evaluate the distribution of the observations using a box plot and a bar plot. We then performed univariate and multivariate logistic regressions and a Wald test. We also calculated the odds ratios and confidence intervals for each variable and the evaluation matrices with receiver operating characteristic (ROC) curves in order to identify cut-off values that are predictive of a diagnosis. Finally, we did the Pearson correlation test to see if the variables grade and IDH were correlated. (3) An excellent ICC estimate was obtained. For the grade and IDH status prediction, there were statistically significant results by evaluation of the degree of post-contrast impregnation (F4) and the percentage of impregnated area (F5), not impregnated area (F6), and necrotic (F7) tissue. These models showed good performances according to the area under the curve (AUC) values (>70%). (4) Specific MRI features can be used to predict the grade and IDH status of gliomas, with important prognostic implications. The standardization and improvement of these data (aim: AUC > 80%) can be used for programming machine learning software.

## 1. Introduction

The most common primary malignant brain tumor is the cerebral glioma, which is a huge family of brain tumors with different characteristics [1]. Previously, the tumor’s grade was determined by the cells’ phenotypic traits [2]. The current World Health Organization (WHO) classification is based on a combination of histology and molecular classification; when differentiating tumors, the two main alterations taken into account are changes in the Isocitrate Dehydrogenase (IDH) gene and the co-deletion of chromosomal arms 1 and 19 (1p/19q) [3]. It might be compared to biomarkers that have an impact on a patient’s prognosis and biological behavior. For instance, regardless of the histological grade, it has been shown that mutations in the IDH gene family provide longer overall survival in high-grade gliomas than their IDH wild-type counterparts [4,5,6].

Moreover, the treatment is influenced by the degree of cellular differentiation and molecular status. For example, according to grading, low-grade gliomas do not typically receive adjuvant radiotherapy and/or chemotherapy. Clinicians discovered that even when patients had the same tumor, their reaction to treatment, degree of side effects, and even prognosis could vary. This suggests that “precision medicine” or clinical therapy based on the needs of specific individuals may represent the way of the future [7,8,9].

Nowadays, immunohistochemical analyses following biopsy or surgical resection represent the most widely utilized technique for determining glioma mutations [10]. Clinical therapeutic planning may be suggested by radiological glioma grading in a non-invasive manner with pertinent prognostic implications [11]. The gold standard for the radiological analysis of gliomas is magnetic resonance imaging (MRI). However, accurate tumor grade identification is far from straightforward because there are not any objective measurements that can be widely replicated and validated [12].

With the aim of standardizing the evaluation of gliomas, the so-called Visually AcceSAble Rembrandt Images (VASARI), developed by multi-institutional neuroradiologists with extensive documented experience in neuro-oncology, take into consideration the fundamental visual characteristics of a standard MRI. The VASARI features are described in detail at https://wiki.nci.nih.gov/display/CIP/VASARI (accessed on 21 March 2020) [13].

The potential of such a model lies in its ability to make objective assessments of tumor characteristics. In fact, although VASARI began as a scale for visual assessments, by calculating the areas of the various tumor components through “regions of interest” (ROIs), it is possible to have a numerical estimate of the features under consideration.

The aim of this paper was to determine which characteristic could be utilized to distinguish between low-grade and high-grade gliomas while attempting to estimate any cut-off values effective for discriminating. Then, we examined the predictive effect of IDH status on the tumor grade and attempted to predict the IDH gene status based on the same morphological parameters.

The main objective of our research is to evaluate the accuracy of the VASARI system in glioma grading and IDH status predictions. This is accomplished by a detailed statistical analysis, which is discussed below and seeks to determine cut-off values for forecasting the data. Lastly, we aim to comprehend the true applicability of these measurements and the VASARI system to machine learning for automated predictions of grade and IDH status with the advent of radiomics through the examination of ROC curves (AUC).

Finally, radiomics has the potential to provide an accurate diagnosis, predict a prognosis, and assess a tumor’s response to therapy [14]. The potential of radiomic analysis to non-invasively distinguish between different glioma molecular subtypes would not only provide additional prognostic information but also help in the selection of targeted chemotherapy in patients with multiple genetic mutations and potentially high-grade tumor types [15,16,17]. It would also help optimize surgeries, on which median survival depends [18]. Thus, radiomic risk models can be used to better predict treatment responses, Progression-free survival (PFS), and Overall survival (OS) [19,20]. By obtaining the radiogenomic profile of a tumor non-invasively, the effect of anti-angiogenic therapies can be assessed without harm to the patient [21,22].

For this purpose, a trio of morphological, textural, and functional signals obtained by the high-throughput extraction of quantitative metrics from voxel-level MR images is used [23,24]. However, because the acquisition parameters have not been standardized and teams have used different methodologies, multicenter studies with different study populations are needed [25,26].

## 2. Materials and Methods

### 2.1. Ethics Statements

The procedure used in this study is not experimental but is ordinarily performed in our hospital; thus, it was approved by the Institutional Review Board. Every patient signed an appropriate, written, informed consent. All data were retrospectively collected. No conflict of interest was manifested by the authors. No funding was received to support this study.

### 2.2. Patients

Our institution’s database was retrospectively examined, and 182 patients who had undergone MRI for pre-surgical glioma evaluation between 2018 and 2021 were identified. Additionally, pathology reports were collected to obtain the glioma grades. Several patients were excluded from the study according to the following criteria: (a) poor acquisition quality imaging; (b) no intravenous contrast; (c) treatments before MR examination, including steroid drugs that may affect edema and contrast enhancement; (d) no pathology reports. Pilocytic astrocytoma (World Health Organization grade 1) was excluded from our study due to its unique imaging characteristics. Finally, 126 patients diagnosed with gliomas were enrolled (Figure 1). The study group comprised 75 males and 51 females, ranging in age from 14 to 84 years (additional data is shown in Table 1).

### 2.3. Magnetic Resonance Imaging Technique

The imaging was carried out at 3.0 T MRI (Magnetom Trio; Siemens Medical Systems, Erlangen, Germany). The MR protocol includes T1-weighted images both before and after the administration of gadolinium-based contrast media, as well as T2-weighted images with dark fluid on the axial planes. We also performed T1-w and T2-w sequences on other planes, as well as DWI and SWI on the axial plane, in addition to these ones. The specific imaging parameters were as follows: (1) axial T1-weighted MR: repetition time of 250 milliseconds, echo time of 2.46 milliseconds, slice thickness of 5 mm, matrix dimensions of 320 × 256, and field of view of 220 × 220 mm^2^; (2) axial T2-weighted MR: repetition time of 6000 milliseconds, echo time of 93 milliseconds, slice thickness of 5 mm, matrix dimensions of 320 × 288, field of view of 198 × 220 mm; (3) axial T2WI dark-fluid MR: repetition time of 8000 milliseconds, echo time of 97 milliseconds, slice thickness of 5 mm, matrix dimensions of 320 × 224, field of view of 181 × 220 mm.

### 2.4. Magnetic Resonance Imaging Assessment and Analysis

According to the VASARI method, we assumed that the entire lesion was composed of the following components: enhancing area, non-enhancing area, necrotic tissue, and edema. Additionally, we extracted the morphological characteristics and the scoring system. Thus, any area of the tumor that exhibits noticeably increased signals on the post-contrast T1-weighted images compared to the pre-contrast was considered an enhancing area. A non-enhancing area was considered any region exhibiting T2-weighted hyperintensity (less than the cerebrospinal fluid intensity) with corresponding T1-weighted hypointensity, along with a mass effect and architectural distortion, including blurring of the gray-white interface.

A portion of the tumor that exhibits necrosis is described as having an uneven boundary, a high signal on the T2-weighted and proton density imaging, and either no enhancement at all or a much lower enhancement. A quantitative assessment of the necrosis was obtained by evaluating the ratio of the area of the total lesion to the area of necrosis (internal to it), as shown in Figure 2. A hemorrhage was identified on the T2-, T1-, and SWI T2*-weighted sequences and evaluated in relation to the presence of hemoglobin degradation products. The diffusion characteristics are defined as predominantly facilitated or restricted in the enhancing or non-contrast enhanced tumor (nCET) portion of the tumor based on an apparent diffusion coefficient (ADC) map. They are defined as mixed in the presence of a relatively equal proportion of facilitated and restricted diffusion.

Edema should be greater in signal than the no-enhancement tumor and somewhat lower in signal than the CerebroSpinal Fluid (CSF). Pseudopods are characteristic of edema. They were scored on the basis of the percentage of total abnormal tissue (0% = 1; <5% = 2; 6–33% = 3; 34–67% = 4; 68–95% = 5; >95% = 6).

The other VASARI features, which were evaluated with dedication and attention, are as follows: tumor location, side of tumor epicenter, eloquent brain areas, enhancement lesion quality, enhancing tumor proportion, non-enhancing tumor (nCET) proportion, necrosis tumor proportion, cysts, multifocal or multicentric aspects, T1/fluid-attenuated inversion recovery ratio, definition of the non-enhancing margins, hemorrhage, thickness of the enhancing margins, definition of the enhancing margins, edema tumor proportion, crossing midline edema, pial invasion, ependymal invasion, cortical involvement, deep white matter invasion, nCET tumor-crossing midline, enhancing tumor-crossing midline, satellites, calvarian remodeling, and size of lesion (diameter).

The imaging features plus a single measurement of the lesion size were valued by 2 residents and 3 neuroradiologists, independently.

### 2.5. Statistical Analysis

The statistical analyses were performed using software (Stata, version 15; StataCorp, College Station, TX, USA). The inter-reader agreement was assessed using the Cohen κ coefficient (Statistical Package for the Social Science Statistics, software version 24; IBM Corp, Armonk, NY, USA). The κ coefficient was interpreted as follows: 0.20 or less, poor agreement; 0.21 to 0.40, fair agreement; 0.41 to 0.6, moderate agreement; 0.61 to 0.80, good agreement; 0.81 to 1, very good agreement.

We built a sub-dataset called “columns” that only contained the following parameters in order to only take into account the relevant columns throughout our evaluations: G (grade), F4 (enhancing quality), F5 (enhancing portion), F6 (non-enhancing portion), F7 (necrosis), etc. We added a new dataset called “levels” to the dataset. This variable is binary, with level = 0 denoting a grade of 1 or 2 (low grade) and level = 1 denoting a grade of 3 or 4 (high grade).

The aim of the statistical analysis was to examine the information to see if we could establish a cut-off point for each covariate (column in the dataset). The objective is to determine if we can categorize the tumor as having a high or low amount of the G variable based on the cut-off. The analyses were performed separately for each variable. We used a box plot and a bar plot to visualize the distribution of the observations for each covariate. The “VASARI” variable’s impact on the prediction of the “degree” response variable was then examined using a univariate logistic regression. In order to find cut-off values that are indicative of the grade diagnosis, we also conducted a Wald test on the categorical variables to establish statistical significance. Additionally, we computed the odds ratios and confidence intervals for each variable and evaluated the matrices with ROC curves. Then, after establishing a cut-off level for each covariate, we reanalyzed the data to see whether we could find a significant correlation between the IDH status and the grade. We created a contingency table with the various grade levels on one side and the IDH status, which was either positive or negative, on the other, and calculated the findings as percentages to determine if there was a correlation between them. To determine whether the variables grade and IDH are connected, we also performed a Pearson correlation test.

## 3. Results

### 3.1. Inter-Reader Agreement

The κ value for the VASARI feature detection was 0.86 (very good agreement; *p* < 0.001).

### 3.2. Grade Prediction

According to the findings of the statistical analysis, not all of the VASARI features have statistically significant differences (*p* < 0.05) between the high- and low-level gliomas. The results that were statistically significant for discerning the glioma grades included the following: F4 (enhancing quality), F5 (enhancing portion), F6 (non-enhancing portion), and F7 (necrosis). The F4 variable has a statistically significant difference between F4 = 2 and F4 = 3 (*p* value = 0.00072 < 0.05) and is statistically significant for the grade prediction (*p* value = 0.00028 < 0.05). In particular, F4 = 3 has a 91% probability of being at the high level (grades 3 and 4), while the probability of the other two levels is much lower (56% and 58%) (Table 2). The model has an accuracy of 0.776, a sensitivity of 0.804, a specificity of 0.652, and an AUC of 0.73.

Additionally, the variable F5 is statistically significant for the grade prediction (*p* value = 0.00388 < 0.05) and shows a cut-off level of F5 = 5. This means that an F5 = 5 has a 91% probability of being at the high level (grades 3 and 4), while the probability of the other levels (F5 < 5) is much lower (56% and 57%). This model has an accuracy of 0.792, a sensitivity of 0.873, a specificity of 0.435, and an AUC of 0.677.

At the same time, variable F6 is statistically significant for the grade prediction (*p* value = 0.0032 < 0.05). The F6 variable has a statistically significant difference between F6 = 3 and F6 = 4 (*p* value = 0.032 < 0.05). In particular, F6 = 3 (or more) has a 92% probability (or more) of being at the high level, while the probability of the other levels (F6 < 3) is much lower (0% and 46%). This model has an accuracy of 0.736, a sensitivity of 0.745, a specificity of 0.696, and an AUC of 0.757 (Figure 3).

Finally, the variable F7 is statistically significant for the grade prediction (*p* value = 0.035 < 0.05) and shows a cut-off level of F7 = 2. For the predicted probabilities of being in the high-level grades, F7 = 3 and F7 = 4 have a probability of 91% and 93%, respectively, of being at the high level. This model has an accuracy of 0.712, a sensitivity of 0.706, a specificity of 0.739, and an AUC of 0.738.

### 3.3. IDH Analysis

A total of 76% of the observations have a negative IDH (non-mutated) and a high grade (G = 3 or 4). The Pearson test (Table 3) gave a *p* value of 2.2 e^−16^ < 0.05; thus, the coefficient is statistically significant. We obtained a correlation coefficient of −0.6978789, meaning that these two variables are significantly negatively correlated. The resulting confidence interval is (−0.7781250, −0.5952169) at 2.5% and 97.5%, respectively.

The F4 values are very different depending on the positive or negative IDH. In fact, a negative IDH corresponds almost 65% of the time to an F4 = 3. On the other hand, only 20% of the observations have a positive IDH, and the F4 levels among them are more or less equally distributed. We found that the variable F4 is significant with a *p* value = 6.78 e^−05^ < 0.05 and that there is a significant difference between F4 = 2 and F4 = 3 (*p* value = 0.0013 < 0.05). An F4 = 3 has a 10% probability of having a positive IDH, while the probability of the other two levels is higher (50% and 42%). This model has an accuracy of 0.776, a sensitivity of 0.640, a specificity of 0.810, and an AUC of 0.73 (Figure 4).

Most of the time, the highest levels of F5 (Figure 5) seem to correspond to a negative IDH, and this variable is statistically significant with a *p* value = 0.00034 < 0.05 and a cut-off level of F5 = 3. Additionally, F5 = 2 and F5 = 3 have a high probability of 50% and 57%, respectively, of having a positive IDH, while the probability of the other levels is much lower (around 15%). This model has an accuracy of 0.808, a sensitivity of 0.48, a specificity of 0.890, and an AUC of 0.73.

The variable F6 is statistically significant with a *p* value = 0.000124 < 0.05 and a cut-off level of F6 = 5. Additionally, F6 = 6 and F6 = 7 have the highest probabilities of 100% and 60%, respectively, of having a positive IDH. While the probability of the other levels is much lower (from 8% to 21.4%). This model has an accuracy of 0.84, a sensitivity of 0.48, a specificity of 0.93, and an AUC of 0.7648.

The variable F7 is statistically significant with a *p* value = 0.00339 < 0.05 and a cut-off level of F7 = 2. This model has an accuracy of 0.744, a sensitivity of 0.800, a specificity of 0.730, and an AUC of 0.789 (Figure 6).

The variable F24 is statistically significant with a *p* value = 0.0367 < 0.05. A total of 77.6% of the observations have F24 = 1, and there is only one observation that has F24 = 2 and a positive IDH. Additionally, F24 = 1 has a 24.7% probability of having a positive IDH. This model has an accuracy of 0.408, a sensitivity of 0.960, a specificity of 0.270, and an AUC of 0.615.

The other variables are not statistically significant.

## 4. Discussion

In our study, we demonstrated with statistical evidence that some VASARI features are indicative of brain glioma grades and IDH statuses. In particular, these features regard the enhancement quality (F4), tumor-enhancing proportion (F5), tumor non-enhancing proportion (F6), and necrosis proportion (F7). We established the cut-off values for some of these specific radiological features. Therefore, we can predict the presence of high-grade gliomas with a good probability in the diagnostic phase or in high-risk patients during follow-ups, improving the clinical management of patients with gliomas.

We also demonstrated the prediction capability of some VASARI features in the IDH gene mutation status with statistical evidence. Finally, according to the recent literature, we found a positive correlation between the IDH mutations and glioma grades.

According to the recent literature, this process could lead to important clinical implications, such as decisions on diagnostic and therapeutic continuations and predictions of overall survival [27,28]. Moreover, this new controlled lexicon (i.e., VASARI) could allow for greater concordance and faster communication between various interlocutors, such as radiologists, surgeons, and oncologists, from different medical teams and different centers.

The correct differentiation of grade 2 and grade 3 gliomas is sometimes challenging. In our study, we found that nCET was a significant factor with the largest AUC (76%). There was a significant difference between F6 = 3 and F6 = 7 (*p* value = 0.0049 < 0.05), F6 = 3 and F6 = 6, F6 = 3 and F6 = 5, and F6 = 3 and F6 = 4 (*p* value = 0.032 < 0.05). For the predicted probabilities to be in the high-level grades at each level, an F6 = 3 (or more) has a 92% probability (or more) of being at the high level, while the probability of the other levels (F6 < 3) is much lower (0% and 46%). These data remind us that measuring the areas of non-enhancing tumors is an important radiological feature to evaluate disease progression.

Gliomas are very heterogeneous; they may consist of an enhancing component that does not always contain anaplastic lesional parts, while the non-enhancing component frequently contains both anaplastic and low-grade lesional parts. The nCET proportion plays an important clinical role in the diagnosis and characterization of an astrocytoma, which is useful in the surveillance and therapeutic monitoring of astrocytomas. The absence of a tumor enhancement (nCET tumor) is a predictor of longer progression-free survival (PFS) and overall survival (OS). In addition, there seems to be a correlation with an IDH1 mutation [28]. However, establishing the nCET proportion of a brain glioma is still very difficult due to its possible heterogeneous nature and undefined margins. First, this is really a false dichotomy in infiltrating gliomas that have “infiltrative edema”, consisting of tumor cells and edema on a brain background. Second, even with advanced methods, including T2 mapping, diffusion tensor imaging, and perfusion imaging, distinguishing a pure edema from a non-enhancing tumor is not easy. Although this conclusion is occasionally disputed, it appears that the rate of necrosis is related to the grade of the glioma and a worse outcome [29,30].

Zhang et al. [31] accurately predicted the IDH genotype in high-grade gliomas using clinical and MRI features in a machine learning algorithm in 120 patients. Similar results were obtained using Visually AcceSAble Rembrandt Images integrated with radiomic features to predict IDH wild-type lower-grade gliomas (II/III) that carry molecular features such as epidermal growth factor receptor (EGFR) amplification or telomerase reverse transcriptase (TERT) promoter mutations, which are reported to behave similarly to glioblastomas [32]. In light of these results, we agree that it could be interesting to propose combining the results that can derive from multiparametric magnetic resonance imaging (MRI) radiomics, qualitative features using the VASARI lexicon, and clinical factors to better understand tumor behavior and tumor classification [33]. Advanced magnetic resonance techniques have shown excellent potential in identifying pathological characteristics of brain tumors useful for their classification. For example, values derived from diffusion-weighted imaging [34] and diffusion kurtosis imaging have allowed for the discrimination between high-grade and low-grade gliomas in some recent studies. IDH wild-type gliomas have been demonstrated to show lower ADC values, which also correlated with a worse prognosis in both IDH mutant and IDH wild-type gliomas, irrespective of their histological grade [35]. Additionally, it has been demonstrated that the mutational status of isocitrate dehydrogenase 1 (IDH1) in anaplastic gliomas can be predicted non-invasively using a texture analysis (TA) of diffusion-weighted imaging (DWI) in combination with conventional magnetic resonance imaging (MRI) [36].

About 10% of glioblastoma multiforme (GM) and 30% of anaplastic astrocytomas may not show enhancement, whereas low-grade gliomas sometimes demonstrate enhancement. Fluid-attenuated inversion recovery (FLAIR) images may depict some features of the tumor but have low specificity. In this setting, 1 proton magnetic resonance spectroscopy (H-MRS) adds value to clinical practices by providing information for the in vivo assessment of the biochemical pathways that contribute to tumor characterization [37,38].

There are still several limitations in our study that should be discussed, which are as follows:

-The classes (low level/high level) are quite unbalanced (23 observations in the low level and 102 in the high level); we are planning to apply a balancing method, such as oversampling or undersampling, to improve the performance of the model.-The recent study showed that glioblastoma patients with a combination of deep white matter tracts and ependymal invasions on the imaging had a significant decrease in their overall survival compared with patients with an absence of such invasive imaging features. In this study, pial and ependymal invasions had a significantly increased risk of high-grade gliomas on the univariate analysis but not on the multivariate regression analysis.-Correlations between the patients’ outcomes and survival times and the VASARI scores were not identified in this study.

That could be an interesting starting point for other studies.

In conclusion, with this study, we have shown that some features of the VASARI are very useful in guiding neuroradiologists toward grade and molecular diagnosis. In particular, we have tested the objectivity of these assessments, and we believe that the results obtained in terms of AUC are satisfactory, although, at the moment, they cannot be used for machine learning processes. From this complex and elaborate statistical analysis, our future goal is to streamline the VASARI score and make it applicable to machine learning processes. For this, we will direct our efforts toward the elimination of features that have proven to be unhelpful for grade and molecular diagnosis and revise the way we assign quantitative variables to the VASARI features that are already capable of predicting grades and IDH statuses with statistical significance. The ultimate goal will be to obtain models with an AUC > 0.8 in order to be used for machine learning.

## Figures and Tables

**Figure 1 jimaging-09-00075-f001:**
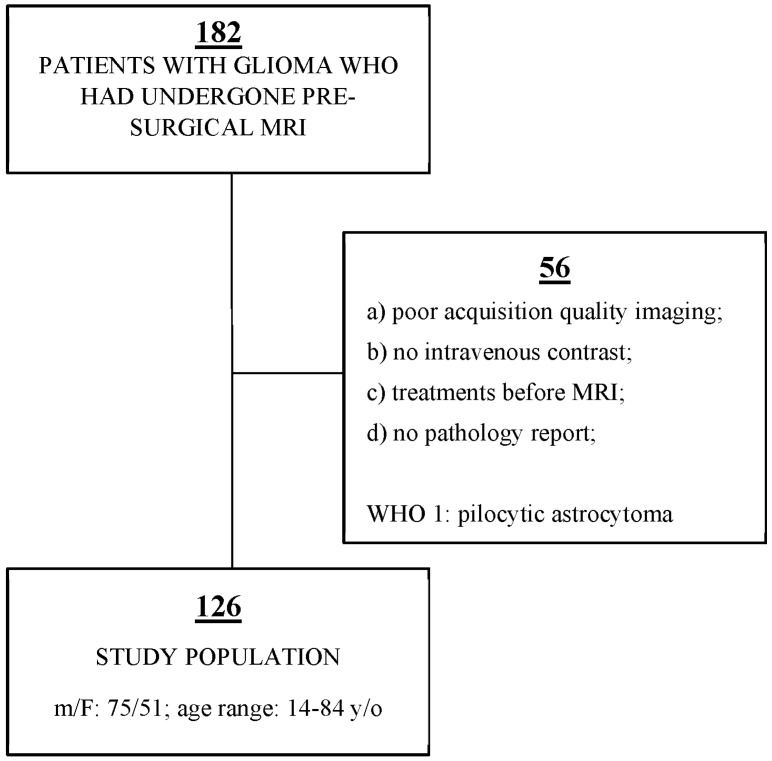
Sample size selected strictly by exclusion criteria (We retrospectively reviewed our institution’s database, identifying 182 patients. We excluded 56 patients due to exclusion criteria. Finally, 126 patients were enrolled in our study. The flowchart shown in the figure also indicates the sex, the age range and the exclusion criteria.).

**Figure 2 jimaging-09-00075-f002:**
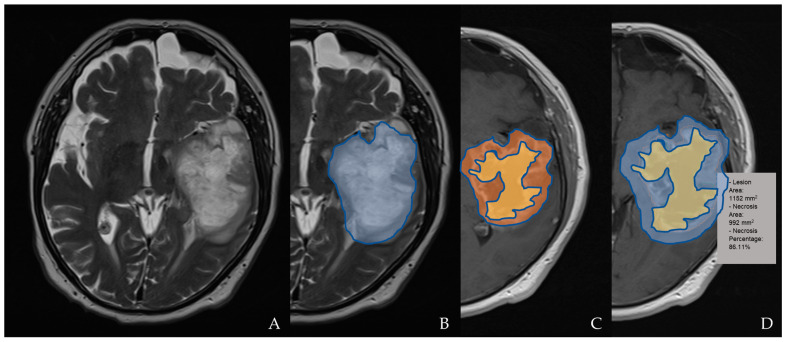
An example of tumor segmentation for necrosis depiction. The segmentation of the T2 hyperintensity region designated as tumor invasion is shown in (**A**,**B**) on an axial plane (blue); the axial post-contrast T1WI shows segmentation of the necrotic area (yellow) compared to the enhancing tumor (orange; in (**C**)) and lesion/tumor area (blue; in (**D**)); the necrosis percentage is also calculated in (**D**).

**Figure 3 jimaging-09-00075-f003:**
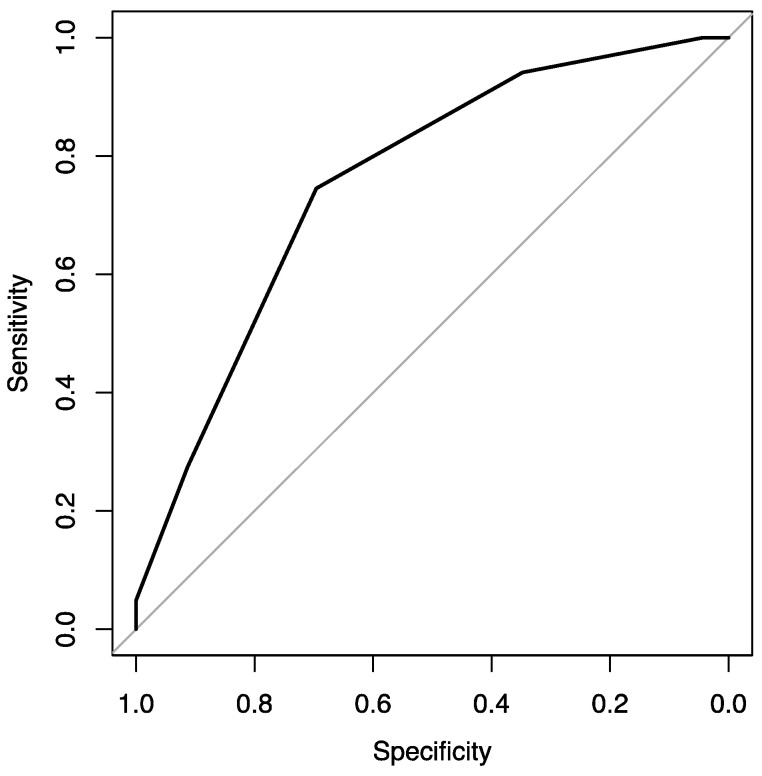
ROC curve showing the F6 parameter’s potentiality in the grade prediction.

**Figure 4 jimaging-09-00075-f004:**
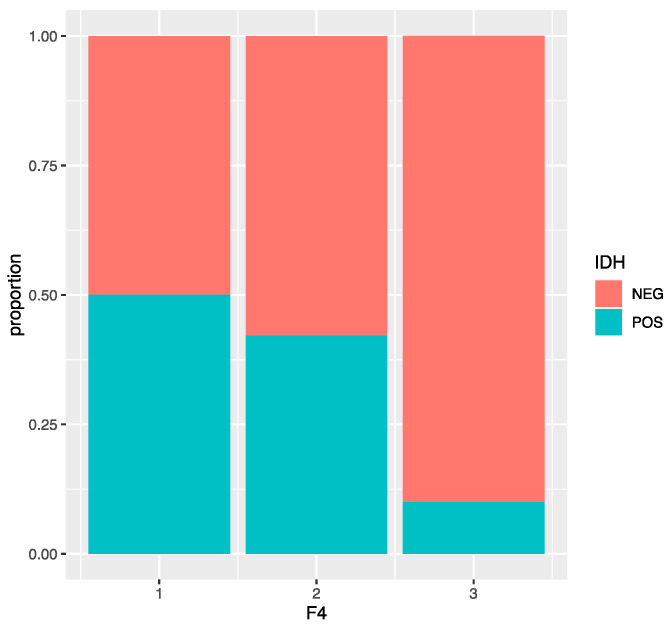
Bar plot representing the distribution of IDH status according to the F4 VASARI variable.

**Figure 5 jimaging-09-00075-f005:**
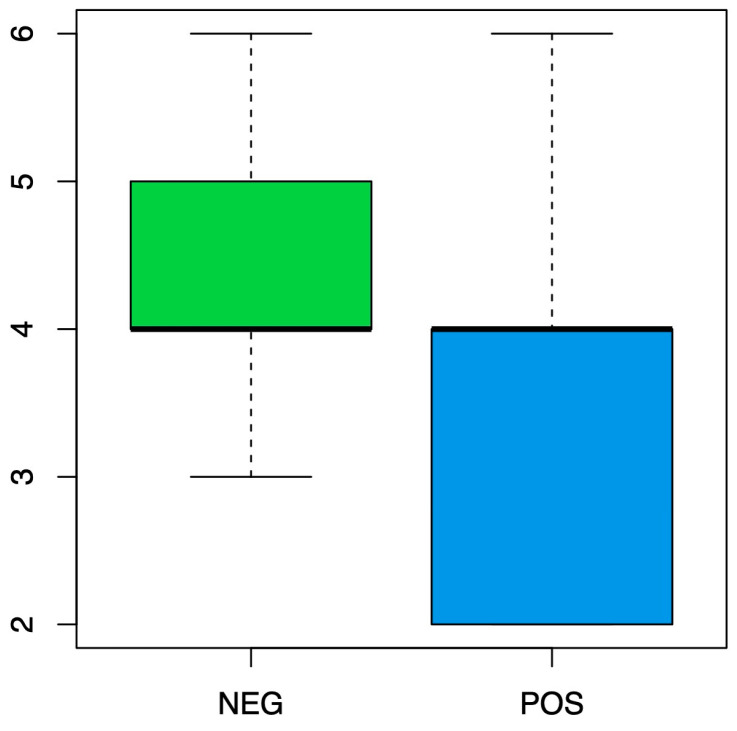
Box plot representing the distribution of IDH status according to the F5 VASARI variable.

**Figure 6 jimaging-09-00075-f006:**
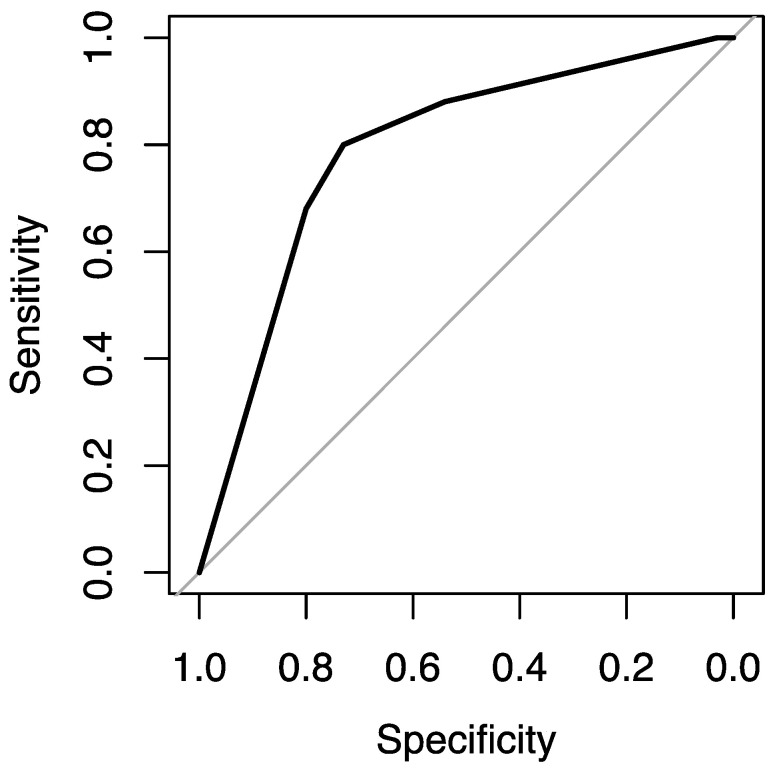
ROC curve showing the F7 parameter’s potentiality in the IDH prediction.

**Table 1 jimaging-09-00075-t001:** Additional demographic data on our study population.

		Glioma Grade				
Other Data		1 (*n* = 3)	2 (*n* = 21)	3 (*n* = 18)	4 (*n* = 84)	
Age (year)	<50	3	10	8	18	39
	>50	0	11	10	66	87
Sex	Male	2	8	11	54	75
	Female	1	13	7	30	51
Location	Frontal	0	12	9	28	49
	Temporal	0	7	4	17	28
	Insular	2	2	1	6	11
	Parietal	0	0	1	22	23
	Occipital	0	0	2	2	4
	Brain steam	1	0	1	5	7
	Other (cerebellum)	0	0	0	4	4
Side	Right	0	11	5	47	63
	Left	2	0	2	5	9
	Central/Bilateral	1	10	11	32	54
Eloquent area	No	2	15	13	45	75
	Motor speech	1	2	1	7	11
	Receptive speech	0	4	2	16	22
	Motor area	0	0	1	15	16
	Visual area	0	0	1	1	2
IDH status	Positive	2	13	3	4	22
	Negative	1	8	15	80	104

**Table 2 jimaging-09-00075-t002:** The corresponding odds ratios and confidence intervals for the F4 parameter are shown in the table. On the right, the predicted probabilities of being in the high-level grade (level 1) are also reported.

F4	OR	2.5%	97.5%	Level 1 Probability
as.factor(F4)1	1.285714	0.4790464	3.59729	0.5625000
as.factor(F4)2	1.069444	0.2748978	4.15047	0.5789474
as.factor(F4)3	7.972222	2.3284132	27.90801	0.9111111

**Table 3 jimaging-09-00075-t003:** Pearson correlation test showing if the variables grade and IDH are correlated.

IDH Status	GRADE				
	1	2	3	4	
Negative	0.8	3.2	10.4	65.6	80.0
Positive	0.8	13.6	4.0	1.6	20.0
	1.6	16.8	14.4	67.2	100

## Data Availability

Data are available upon reasonable request to the corresponding author.

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
