# Peer review of "Vasari Scoring System in Discerning between Different Degrees of Glioma and IDH Status Prediction: A Possible Machine Learning Application?"

_2313-433X, 2023, doi:10.3390/jimaging9040075_

Round 1

Reviewer 1 Report

The authors present a manuscript of clinical interest. The subject falls within the scope of the journal. Description and discussion of the findings well done and well-founded. Overall, the paper is well written and contains valuable information. The bibliography is pertinent and current. However, the text still needs some improvement and minor repairs.  The potential of such a model lies in its ability to make objective assessments of tumor characteristics. In fact, although VASARI

Did you quantify necrosis? How?

In which sequence hemorrhage was defined?

Diffusion characteristics?

Show MRI findings of some cases to illustrate.

2 residents and 3 neuroradiologists ? Did you think the residents had the same performance as neuroradiologists? This was not considered in discussion.

INCLUDE THE FOLLOWING REFRENCE IN THE DISCUSSION

Messina C, Bignone R, Bruno A, Bruno A, Bruno F, Calandri M, Caruso D, Coppolino P, Robertis R, Gentili F, Grazzini I, Natella R, Scalise P, Barile A, Grassi R, Albano D. Diffusion-Weighted Imaging in Oncology: An Update. Cancers (Basel). 2020 Jun 8;12(6):1493.

include:

About 10% of GM and 30% of anaplastic astrocytomas may not show enhancement, whereas low-grade gliomas sometimes demonstrate enhancement. FLAIR images may depict some features of the tumor but have low specificity. In this setting, 1 H-MRS adds value in clinical practice, adding information for the in vivo assessment of biochemical pathways that contribute to tumor characterization

Durmo F, Rydelius A, Baena SC, et al. Multivoxel 1 H-MR spectroscopy biometrics for preoperative differentiation between brain tumors. Tomography 2018; 4(4): 172–181

Farche MK, et al. Revisiting the use of proton magnetic resonance spectroscopy in distinguishing between primary and secondary malignant tumors of the central nervous system. Neuroradiol J. 2022 Oct;35(5):619-626. 

Author Response

We appreciate the reviewer's insightful comments. Listed below are our responses.

Reviewer 2 Report

The authors analyze the stability and predictive ability of VASARI features for predicting glioma grade and IDH mutation from an MRI.

1. The authors claim in Introduction (on line 70) that VASARI scores are objective, however, they are determined by radiological assessment. Were there any biases observed by the authors while extracting the VASARI scores?

2. The list of VASARI features that has been presented in Introduction - details like this should be moved to the Methods section of the paper.

3. In Table 1, the authors provide don't provide study cohort characteristics except for age and sex. It is important to show the distribution of glioma grades and IDH mutations for the analysis cohort. It is important to report other well-known confounders for predicting IDH mutation status and glioma grade.

4. In statistical analysis section, lines 149 - 154, why did the authors create a subset of the data? What was the rationale for selecting this subset? Was it the univariate logistic regression? If so, this should be mentioned earlier in the statistical analysis section.

5. The authors state kappa values for skull fracture detection (in line 174) - why did the authors report skull fracture detection? Was it one of the goals of this study? This needs to be clarified.

6. A pertinent problem with this paper is that it fails to cite and compare results with previous studies that develop models for predicting IDH mutation status and glioma grade. For instance, authors should compare their results with:

Zhou, Hao, et al. "MRI features predict survival and molecular markers in diffuse lower-grade gliomas." Neuro-oncology 19.6 (2017): 862-870

6b. Su, C-Q., et al. "Combined texture analysis of diffusion-weighted imaging with conventional MRI for non-invasive assessment of IDH1 mutation in anaplastic gliomas." Clinical Radiology 74.2 (2019): 154-160.

7. In recent years, radiomics-based approaches have been demonstrated to be particularly effective at predicting glioma grades and IDH mutations. However, none of the studies have been cited in this study. The authors should include a section on radiomics in the Introduction and compare how the VASARI scores are more effective than radiomics based approaches?

8. What validation scheme was used by the authors for computing the AUCs? The authors should report results either from 10-fold cross validation or bootstrapping.

9. The figures representing AUCs need to be visually improved by adding lines for random choice and different AUCs for 10-fold cross validation

10. The two different AUCs can be grouped into a single figure with different panels. 

11. The authors state that one of the major goals of this study were to evaluate the stability of VASARI features - yet no results are comprehensively presented.

Author Response

(The authors gave the same response as above.)

Reviewer 3 Report

Thank you for the opportunity to review this paper. Please find my comments below.

The paper is very well written, and the author had clear guidelines to follow proper methodology to conduct their study.

-          I have a concern about which data were used to train the ML before conducting the study.

-          What is the motivation to conduct the study, and why did the authors select Glioma?

-          How did the authors obtain patient consent for the retrospective study?

-          Is there any previous CT conducted for the patients, and how can it change the study results?

-          How do the authors justify that the sample size (126) is enough to come up with the results and recommendation?

-          The author should clarify the suitability of other machines or protocols to come up with the same results.

Author Response

(The authors gave the same response as above.)

Round 2

Reviewer 2 Report

No further comments.